# Effect of Fortified Inoculation with Indigenous *Lactobacillus brevis* on Solid-State Fermentation of Light-Flavor *Baijiu*

**DOI:** 10.3390/foods12234198

**Published:** 2023-11-21

**Authors:** Xiaoxue Chen, Xiaoning Huang, Shanfeng Sun, Beizhong Han

**Affiliations:** Key Laboratory of Precision Nutrition and Food Quality, Department of Nutrition and Health, China Agricultural University, Beijing 100089, China; chen.xx@cau.edu.cn (X.C.); hxning926@sina.com (X.H.); 18302432883@163.com (S.S.)

**Keywords:** light-flavor *Baijiu*, *Lactobacillus brevis*, solid-state fermentation

## Abstract

*Baijiu*, one of the world’s oldest distilled liquors, is widely consumed globally and has gained increasing popularity in East Asia. However, a comprehensive understanding of the underlying principles behind this traditional liquor product remains elusive. Currently, *Baijiu* is facing the industrial challenge of modernization and standardization, particularly in terms of food quality, safety, and sustainability. The current study selected a *Lactobacillus brevis* strain based on experiments conducted to assess its environmental tolerance, enzyme activity, and fermentation performance, and highlight its exceptional fermentation characteristics. The subsequent analysis focused on examining the effects of fortifying the fermentation process of *L.brevis* on key microbiotas, physicochemical parameters, and volatile profiles. The qPCR results revealed that the inoculated *L. brevis* strategically influenced the the composition of the dominant microbial communities by promoting mutual exclusion, ultimately leading to improved controllability of the fermentation process. Moreover, the metabolism of the inoculated *L. brevis* provided more compounds for the formation of flavor profiles during fermentation (the content of ethyl acetate was increased to 57.76 mg/kg), leading to a reduction in fermentation time (from 28 d to 21 d). These findings indicate promising potential for the application of the indigenous strain in *Baijiu* production.

## 1. Introduction

*Baijiu*, a traditional fermented alcoholic drink that originated in China, is typically produced from natural fermentation. It is highly regarded and plays a significant role in the Chinese culinary culture [1,2]. *Baijiu*’s flavor profile allows for its classification into three main categories: Sauce-flavor *Baijiu*, Strong-flavor *Baijiu*, and Light-flavor *Baijiu*. Additionally, it can be further classified into nine subgroups: miscellaneous-flavor *Baijiu*, Feng-flavor *Baijiu*, Rice-flavor *Baijiu*, Medicine-flavor *Baijiu*, Sesame-flavor *Baijiu*, Te-flavor *Baijiu*, Chi-flavor *Baijiu*, Laobaigan-flavor *Baijiu*, and Fuyu-flavor *Baijiu*. Among these categories, the three main categories hold a dominant position in the *Baijiu* market, collectively accounting for approximately 60% to 70% of the *Baijiu* consumption in China [1].

*Fenjiu*, a notable example of Light-flavor *Baijiu*, is primarily manufactured in Northern China using sorghum as the raw material. Alcoholic fermentation is conducted by combining low-temperature *Daqu*, derived from barley and pea, with sorghum [1]. The production process of *Fenjiu* consists of two rounds of fermentation [3]. This study centers on the first-round fermentation, which usually lasts for 28 days. It can be further divided into three primary main stages: saccharification stage, main fermentation stage, and flavor production stage [4]. The solid-state fermentation process of *Fenjiu* entails a unique and intricate simultaneous combination of saccharification and spontaneous fermentation [5]. This phase typically becomes active on the seventh day of the fermentation process. Subsequently, the fermentation process progresses into the main flavor production stage, which lasts for 21 days, but there is potential for shortening this duration, For instance, fortified fermentation with Bacillus elicits a marked reduction in the duration required to attain peak fermentation temperature [6]. Meanwhile, the application of multi-omics technology enables the discernment of core functional microbes and phages, thereby affording the potential to expedite the fermentation process and enhance its overall efficiency [7].

Traditionally, the starter for *Fenjiu* fermentation was produced in an open-air environment using non-autoclaved raw materials [8,9]. This approach could yield *Daqu* products with inconsistent quality, resulting in an unpredictable and uncontrollable fermentation process for *Fenjiu*. Consequently, these practices may result in significant quality defects and instability in the *Baijiu* product, even when the final product undergoes blending [10]. Therefore, conducting a comprehensive study on the fermentation process of *Baijiu* is essential for transitioning from poorly controlled spontaneous fermentation to a precision fermentation approach, employing specific strains under strict process control.

Spontaneous solid-state fermentation processes undergo various dynamic changes, including microbial growth, glucose and oxygen consumption, metabolite products, temperature changes, and moisture loss. Every one of these factors plays a critical role in determining the quality and productivity of the fermentation. However, controlling the environmental parameters that influence spontaneous fermentation is generally challenging. As a result, inoculated fermentation has become a powerful practice in the food industry due to the enhanced control it offers [11,12,13]. The prevalent use of selected starters accelerates and guides the fermentation process while enabling quality prediction of the final products. Importantly, during fermentation, the dynamics of microbiota were significantly altered when inoculated with selected starters compared to the uninoculated control group [10]. Moreover, inoculating specific strains can influence the metabolic activity of the microbial community and regulate the flavor profiles of fermented foods. For example, when selecting *Leu. mesenteroides* for spontaneous kimchi fermentation, there was an increase in *Leuconostoc* proportions and a decrease in *Lactobacillus*, whereas the use of a starter culture was deemed for producing standardized kimchi with consistently high quality [14]. Furthermore, using *S. cerevisiae* and *Le. mesenteroides* as selected starters in the fermentation of traditional strawberry vinegar and kimchi led to a substantial decrease in fermentation time [14,15]. Similarly, in beer fermentation, the use of *C. fabianii* inoculated together with traditional brewers’ yeast (*S. cerevisiae*) was demonstrated to customize aroma profiles and the ultimate ethanol content of beer [16]. In *Baijiu* fermentation, the introduction of *Bacillus* and *Lactobacillus* through fortified fermentation exerts discernible influences on the compositional structure of the microbiome throughout the fermentation process and enhances the flavor and quality of *Daqu* or *Baijiu* [17].

Yeast and Lactobacillus represent pivotal microbial constituents within the fermented grains during the *Baijiu* fermentation process [18]. Substantiated evidence indicates that fortified fermentation possesses the capacity to induce modifications in the native microbial composition, resulting in discernible improvements in the quality and flavor profiles of the final product. In our pursuit of enhancing the fermentation process for light-flavor *Baijiu* production, we selected an indigenous *Lactobacillus brevis* strain with exceptional fermentation characteristics for fortified inoculation, aiming to accelerate the fermentation process and improve the control of the solid-state fermentation. To the best of our knowledge, this report represents the first study on fortified inoculation for the solid-state fermentation process of light-flavor *Baijiu*, utilizing the indigenous *L. brevis*. The microbial content and physicochemical parameters in the fermentation process after inoculation were analyzed, and the volatile components were analyzed using headspace solid-phase microextraction coupled with gas chromatography-mass spectrometry (HS-SPME-GC-MS).

## 2. Materials and Methods

### 2.1. Isolation and Identification of Bacteria and Yeasts

Fermented grain samples were collected from a famous Distillery Co. in Fenyang, Shanxi, China. Each sample (10 g) was suspended in 90 mL of phosphate-buffered saline (PBS) solution, and was homogenized, followed by 10-fold serial dilutions, and spread on MRS agar (Oxoid Co., Hampshire, UK) with 500 μg/mL natamycin, as well as the yeast extract peptone dextrose (YPD) agar medium containing 100 g/L chloramphenicol (Sigma Aldrich Co., St. Louis, MO, USA). The plates were incubated at 30 °C for 24–48 h. Further sub-culturing of single colonies was performed to obtain distinct individual pure cultures for identification [19]. The genomic DNA of the isolates was extracted using bacterial and fungal DNA extraction kits (Tiangen Biotech, Inc., Beijing, China). The species were identified using Sanger sequencing of the 16S and 26S rRNA gene regions. For bacteria, the universal primers B-f (5′-AGAGTTTAGTCCTGGCTCAG-3′) and B-r (5′- AAGGAGGTGATCCAGCCGCA-3′) were utilized [12], whereas for yeasts, NL-1 (5′-GCATATCAATAAGCGGAGGAAAAG-3′) and NL-4 (5′-GGTCCGTGTTTCAAGACGG-3′) were employed [20].

### 2.2. Environmental Tolerance of Various Species

The strains were cultured in YPD (for yeasts) and MRS broth (for LAB) until reaching an OD_600_ of 1 ± 0.2. Subsequently, the environmental tolerance of each strain was assessed using the 96-well cell culture plate. For the ethanol tolerance test, each strain was inoculated (3% *v*/*v*) into MRS or YPD broth supplemented with 10% or 12% (*v*/*v*) ethanol and incubated at 30 °C for 24–48 h. For the pH tolerance test, each strain was also inoculated (3% *v*/*v*) into MRS or YPD broth adjusted to pH 3.0 or 4.0 with 1 M HCl and incubated at 30 °C for 24–48 h. The tolerance ability was determined by measuring the OD_600_ value using a microplate reader (Varioskan Flash, Thermo Fisher Scientific Inc., Waltham, MA, USA).

### 2.3. Enzyme Activities in Various Species

The strains were cultured in YPD (for yeasts) and MRS broth (for LAB) in a 50 mL flask at 30 °C for 7 d. The cultures were centrifuged at 10,000× *g* for 10 min at 4 °C, and the supernatant was collected for exo-enzymatic activity [21], including esterase, amylase, and protease activity. Esterase activity was determined using the p-nitrophenyl (pNP) method with 10 mM 4-Nitropheyl acetate as the substrate [22]. One unit of esterase activity was defined as the amount of enzyme that released 1 μmol of p-nitrophenyl-palmitate (pNP-palmitate). Amylase activity was measured using the DNS method with 1% (*w*/*v*) starch as the substrate [23]. One unit of amylase activity was defined as the amount of enzyme that liberates 1 mmol of reducing sugars per min. Protease activity was assayed using the Folin–Ciocalteu method with 1% (*w*/*v*) casein as the substrate [24]. One unit of protease activity was defined as the amount of enzyme liberating 1 μg of tyrosine per minute.

### 2.4. Fortified Fermentation Inoculated with Promising Candidates

Based on the characterization of all the LAB and yeast isolates, four strains, namely *L. brevis* (Lbr9, Lbr17), *Sa. fibuligera* (Sf6), and *S. cerevisiae* (Sc12) were selected for lab-scale fortified fermentation. Each strain was cultivated in 10 mL of MRS (for LAB) or YPD (for yeasts) broth at 30 °C for 24–48 h and then washed twice using sorghum hydrolysate medium. The cultures of each selected strain were adjusted to ~10^8^ CFU/mL and inoculated to each fortified group for a 28-day simulated solid-state fermentation of light-flavor *Baijiu* [12]. In brief, initially, the fresh sorghum underwent a process of hydration using distilled water. Subsequently, it was subjected to steaming and cooling to room temperature. Following this, *Daqu* was blended with the selected microbial strains. Each individual strain culture, along with *Daqu*, was uniformly integrated into the composite mixture. This composite blend was then subjected to fermentation in airtight bottles for a duration of 28 days. The mixture that had been inoculated with the selected strains was designated as the fermentation group (T group). In contrast, the mixture solely containing *Daqu*, without the introduction of microbial inoculation, was referred to as the control group (CK group) [1].

### 2.5. Analysis of Volatile Compounds in Fermented Grains

The volatile compounds were extracted and analyzed from fermented grains following the method described by Pang et al. [8]. Briefly, a 2 g sample was added to a centrifuge tube containing 8 mL of sterile ultrapure water. After subjecting the sample to ultrasonic wave treatment for 30 min, it was centrifuged at 8000× *g* and 4 °C for 10 min. Then, 8 mL of aqueous extracts, 2 μL of internal standard (125.0 mg/L 4-methyl-2-pentanol), and 3 g of NaCl were introduced into a 20 mL vial. Extraction was performed using SPME fiber (PDMS/CAR/DVB, Supelco, Bellefonte, PA, USA) sampling at 50 °C for 45 min, and then GC-MS (6890-5975B, Agilent, Santa Clara, CA, USA) equipped with an HP-INNOWAX (Agilent, USA) capillary column was aused to isolate and detect volatile compounds in the sample. The heating procedure of the oven temperature was as follows: it was initially heated at 50 °C for 2 min, and then gradually heated at a rate of 2 °C per minute until it reached 85 °C for 0.1 min, and finally, it was heated at a rate of 5 °C per minute until it reached 230 °C for 2 min. The ion source temperature was 230 °C, the ionization mode was EI, the electron energy was set at 70 eV, the four-stage rod temperature was 150 °C, and the mass scanning range was set at 20–350 u.

### 2.6. Quantifying Microbial Dynamics during Fermentation

Fermented grain samples (~150 g) were collected at 0, 1, 3, 7, 15, 21, 25, and 28 days during fortified fermentation. To quantify the abundances of dominant groups of bacteria and fungi, five individual qPCR assays based on a standard curve were conducted using a Fluorescent Quantitative PCR Detection system (Bioer, Hangzhou, China) with a commercial kit (FastStart Essential DNA Green Master, Roche, Mannheim, Germany) [25]. Absolute quantification of each target gene required a standard curve, which can be constructed using genomic DNA from recombinant plasmids carrying the target gene insert. The details on primer sequences and thermal cycles are given in Appendix A. The correlation coefficient (R^2^) for the amplification of specific genes of bacteria, fungi, *Lactobacillus* spp., *Saccharomyces* spp., and *L. brevis* Lbr17 were 0.995, 0.999, 0.998, 0.999, and 0.997, respectively. The microbial metagenomic DNA in fermented grains was extracted using the TIANamp Soil DNA Kit (TIANGEN BIOTECH, Beijing, China). Each 20 μL reaction contained the following: 10 μL Master mix (Roche, Mannheim, Germany), 1 μL of each primer (10 mM), 5 μL of ddH_2_O, and 5 μL of DNA template. Melting curve analysis and agarose gel electrophoresis confirmed the specificity of PCR amplification.

### 2.7. Physicochemical Properties of Fermented Grains

The pH and titratable acidity content of fermented grains were determined according to the method described by Pang et al. [8]. Moisture content was determined using a standard oven drying method at 105 °C until a constant weight was achieved. The reducing sugar content was determined using a colorimetric method with 3,5-dinitrosalicylic acid [26].

### 2.8. Statistical Analysis

All the experiments were performed three times. One-way ANOVA was conducted to determine any significant difference between the samples, with a statistical significance level set at *p* < 0.05. The data used for the heatmap were normalized by row and visualized using TBtools. HS-SPME-GC-MS data were analyzed using Principal Component Analysis (PCA) based on the ‘stats’ and ‘vegan’ packages of the R software (4.3.2). qPCR dataset was log-transformed.

## 3. Results

### 3.1. Isolation and Identification of LAB and Yeasts from Fermented Grains

A total of 76 strains were obtained from fermented grains, including 50 LAB isolates (21 *L. brevis*, 14 *L. hilgardii*, 5 *L. buchneri*, 5 *L. casei* and 5 *L. paracasei*), and 26 yeast isolates (19 isolates of *S. cerevisiae*, 3 *H. osmophila*, 2 *Sa. fibuligera*, 1 isolates *P. fermentans* and 1 *W*. *anomalus*) (Appendix A).

### 3.2. Characterization and Screening of Isolated Strains

To evaluate the potential candidates that could contribute to fermentation, their environmental tolerance and capability of certain enzyme production were tested. A total of 76 isolates was among the LAB isolates; *L. buchneri* (Lbu1, Lbu3), *L. brevis* (Lbr8, Lbr9, Lbr23), *L. casei* (Lca27, Lca28, Lca29), *L. paracasei* (Lpa34, Lpa35, Lpa36), and *L. hilgardii* (Lhi46, Lhi47) exhibited better ethanol tolerance. Most LAB strains showed relatively poor tolerance to pH 3, whereas the majority of them could survive in a pH 4 environment. Regarding yeast strains, most of them displayed strong tolerance to the environment, except for *Sa. fibuligera* (Sf6). *P. fermentans* (Pf1), *Sa. fibuligera* (Sf5), *S. cerevisiae* (Sc8, Sc20), and *W. anomalus* (Wa26) showed strong tolerance to both pH and ethanol. Generally, LAB strains exhibited more noticeable intraspecific phenotypic diversity compared to yeasts (Figure 1).

In terms of enzyme activities, LAB Lbr9 showed the highest protease, amylase, and esterase activity, with values of 12.70, 0.25, and 2329.83 U/mL. Lbr17 also exhibited relatively high esterase activity (2316.67 U/mL). Most LAB strains displayed low amylase activities, ranging from 0.11 to 0.24 U/mL. Among yeast isolates, Sf6 was predominant in enzyme production, with the strongest enzyme activities. The highest protease, amylase, and esterase activities observed were 12.82, 2.38, and 59.4 U/mL, respectively. Additionally, Sc12 showed relatively high esterase activity with a value of 27.98 U/mL (Figure 2).

Therefore, Lbr9, Lbr17, Sf6, and Sc12 were selected as candidate strains.

### 3.3. Fortified Fermentation with Promising Candidate Strains

To assess the contribution of promising candidate strains to simulated *Baijiu* fermentation, two LAB stains (Lbr9 and Lbr17) and two yeast stains (Sf6 and Sc12) were individually employed for fortified fermentation. The fortified groups were characterized based on volatile compounds using PCA analysis (Figure 3). From the score plot, all the fortified groups were differentiated from the control group. Trial Sc12 and Sf6 showed similar results. Fourteen volatile compounds were identified as key compounds responsible for the variations observed in different fermentation trials (Figure 3B). Lbr17 and CK group fermentation exhibited a positive correlation with ethyl acetate and 3-methyl-1-butanol acetate production, and a negative correlation with acetic acid 2-hydroxy-propanoic acid ethy ester, 2-nonanol, and isoamyl lactate production. Sc12 and Sf6 fermentation exhibited a positive correlation with isoamyl lactate, 2-nonanol, and 3-methyl-1-butanol acetate production. Lbr9 was positively related to benzyl alcohol. The volatile compound profile of fortified fermentation with Lbr17 not only contributed to ethyl acetate production but also showed a similar volatile compound profile with the CK group compared to other fermentation trials. Therefore, Lbr17 was selected to simulate a fermentation experiment to study the effect of enhanced inoculation on the fermentation process.

### 3.4. Effect of Fortified Inoculation with L. brevis Lbr17 on the Fermentation Process

In order to interpret the effect of fortified inoculation with Lbr17 on the fermentation process, we monitored the dynamics of biomass for total bacteria, total fungi, *Lactobacillus*, *Saccharomyces*, and *L. brevis* throughout the fermentation process, as depicted in Figure 4. Additionally, we examined the concentration of the main volatile compounds after fermentation.

Total bacteria: As displayed in Figure 4A, the change in total bacteria during the pre-fermentation period was similar between the fortified fermentation group (T group) and the control group (CK Group). However, after inoculation on the 25th day, the total bacteria count in the fortified fermentation group (T group) increased significantly. Importantly, the increase in the total bacteria population was not attributed to changes in the number of *Lactobacillus*, as shown in Figure 4C. In addition to *Lactobacillus*, other bacteria involved in the fermentation process of fermented grains include *Pediococcus*, *Streptomyces*, *Bacillus*, *Leuconostoc*, *Microbacterium*, and *Corynebactrium* [8]. Therefore, the rapid increase in bacterial count after inoculation on the 25th day may be attributed to the reproduction of one or more of these bacteria. However, it is worth noting that this change in bacterial count did not have a noticeable impact on the pH and titratable acidity, suggesting that it would not lead to rancidity.

Total fungi: As shown in Figure 4B, the copy number of total fungi exhibited a similar overall trend in both the fortified fermentation group (T group) and the control group (CK group). Initially, all of them showed a decrease on the first day, followed by an increase. This is likely due to the inoculation of the microbial community from *Daqu* into the grain, leading to a decrease in the total fungal count as the fungi were not yet adapted to the new environment. Subsequently, as the fungi adapted to the environment, there was an increase in the copy number of the fungal population. However, there are differences between the two groups. In the fortified fermentation group (T group), the total number of fungi decreased sharply from the 15th to the 21st day, reaching its lowest value on the 21st day, but subsequently recovered. This is possibly due to the accumulation of acid produced by inoculated *L. brevis*, which leads to the death of fungi that cannot tolerate higher acidity. The recovery may be attributed to the reproduction of fungi that can tolerate higher acidity, filling the niche left by the dead fungi. Throughout the entire fermentation process, the total number of fungi in the fortified fermentation group (T group) was lower than that in the CK group. This indicates that the inoculated *L. brevis* (Lbr17) had an inhibitory effect on the growth of fungi during the fermentation process.

*Lactobacillus*: Figure 4C,E clearly demonstrate that the inoculation of *L. brevis* has a significant impact on the number of *Lactobacillus* during the fermentation process. During the initial 15 days, the fortified fermentation group (T group) exhibited a higher number of *L. brevis* compared to the control group (CK group) due to the initial abundance of *L. brevis*. The relationship between *L. brevis* and other *Lactobacillus* exhibited mutually exclusive effects, whereas the extensive reproduction of *L. brevis* hindered the growth of other *Lactobacillus* strains. Beyond the 15-day mark, the total number of *L. brevis* in the fortified fermentation group (T group) decreased compared to the control group (CK group), which can be attributed to a decrease in the overall number of *L. populations*. This finding holds significant value in guiding targeted adjustments to unexpected microorganisms during the fermentation process through the use of mutually exclusive microorganisms. Furthermore, it reveals how fortified inoculation enhances the controllability of the fermentation process.

Yeast: The overall trend of yeast during fermentation is consistent with the total fungal trend. Interestingly, the number of yeasts reached its lowest value at 21 d, indicating a consistent response of total fungi and yeast to the fermentation environment changes following the fortified inoculation of *L. brevis* (Lbr17). This finding serves as a reference for determining the inoculation time of fortified fermentation with *L. brevis* (Lbr17).

*L. brevis*: During the fermentation process, *L. brevis* exhibited dynamic changes. The direct impact of fortifying the inoculation of *L. brevis* was observed only within the first 15 d, with the fortified fermentation group (T group) showing a higher number of *L. brevis* compared to the control group (CK group). Afterwards, the copy number of *L. brevis* in both groups tended to stabilize. From days 25 to 28, there was a slight decrease in the number of *L. brevis*, whereas the number of other bacteria in the *Lactobacillus* genus remained stable. However, the total bacterial count increased during this period. This could be attributed to *L. brevis* promoting the growth of bacteria not belonging to the *Lactobacillus*, whereas its own growth displayed a declining trend, possibly due to limitations in spatial niche availability.

The fermentation trials showed similar initial pH and titrate acid values (Figure 5A). Both pH and titratable acid exhibited more significant changes within the first 7 days, followed by a relatively stable trend. On the first day, there was a sharp change in pH, decreasing from 6.2 to 4.1, and an increase in titratable acid from 0.06 to 0.12 mmol NaOH/g. The T group showed a lower pH value and higher titrate acid value as compared to the CK group after the first day of fermentation. The moisture content and reducing sugar levels during the 28-day fermentation process are depicted in Figure 5B. The overall trend for reducing sugar and moisture content was consistent. Both reducing sugar and moisture content decreased on the first day, followed by a slight increase on the third day, and then rapidly declined. After inoculation on the 15th day, they tended to stabilize. When comparing the fortified fermentation group (T group) to the CK group, the moisture content and reducing sugar levels of the fortified fermentation group remained at a higher level, indicating a more stable fermentation process.

The volatile compounds present at 21 d and 28 d during fermentation were analyzed using HS-SPME-GC-MS (Figure 6). When comparing the main volatile flavors of the fortified fermentation group (T group) (Lbr17) at 21 days and the control group (CK group) at 28 days, it was found that the content of main flavor compounds, such as isoamyl acetate and 1-hexanol increased significantly without significant decrease in the ethanol concentration at 21 d in the T group (Lbr17). Additionally, some important volatile flavor compounds, including ethyl acetate, 2-methoxy-phenol, diethyl suberate, 2,4-di-tert-butylphenol, benzyl alcohol, and ethyl octanate, showed slight increases.

## 4. Discussion

The growth of LAB and yeast populations is influenced by their diverse ability to utilize sugars and amino acids, and their tolerance to various stresses, including acidity and high concentrations of ethanol [27]. In the late stage of fermentation, where the ethanol content in fermented grains exceeds 10% and the pH can drop as low as 3.5, strains that are unable to tolerate these conditions will cease to grow or even die. Therefore, ethanol and pH tolerance are critical indicators for screening functional bacteria in liquor fermentation. To assess the strain’s adaptability to the fermentation environment, its ethanol and pH tolerance were tested.

Light-flavor *Baijiu* is characterized by the presence of ethyl acetate and ethyl lactate. Yeast strains are capable of producing esters, particularly ethyl acetate. LAB have the ability to produce lactic acid and form ethyl lactate through the action of esterases [13,28,29]. In our study, the Lbr8 and Lbr9 isolates showed ethanol tolerance. Some strains of *L. buchneri* and *L. hilgardii* (Lbu3 and Lhi46) showed lower ethanol tolerance abilities than *L. brevis*, but still exhibited tolerance to 12% ethanol. Generally, *L. brevis*, *L. buchneri,* and *L. hilgardii* demonstrated slightly higher ethanol tolerance compared to other LAB strains. This suggests that these three species may be important constituents of the microbial community during the late stage of fermentation. The ethanol tolerance of bacteria is influenced by several factors, including ethanol-induced changes in plasma membrane composition and inactivation of cytosolic enzymes (such as ATPase and glycolytic enzymes) [30]. Yang et al. reported that increased stress response and fatty acid biosynthesis, along with decreased amino acid transport and metabolism, may play important roles in strains’ adaptation to environmental ethanol [31]. Some *L. brevis* strains (Lbr8 and Lbr9) showed a high ethanol tolerance, possibly linked to the formation of cell macro-fibers and structured filamentous growth in response to ethanol stress [32]. LAB and yeasts are regarded as functional microbiota responsible for alcohol and flavor compound production. In our study, 4 out of 50 LAB strains and 5 out of 26 yeasts showed superior environmental tolerance compared to others. Enzyme activity was then determined for these selected LAB and yeast.

Only one strain of LAB (Lbr9) out of the four strains with better tolerance exhibited preferable enzyme activity. Since esterase has a crucial role in the formation of flavor compounds, Lbr17 was selected based on its high esterase activity. As a result, two LAB (Lbr9, Lbr17) were chosen for the fermentation test. For yeasts, the activity of enzymes in five strains with good tolerance was poor. Therefore, Sf6, which displayed good activity in three enzymes, was selected as the yeast strain. Considering the important role of esterase in the formation of flavor compounds, Sc12 was chosen for its excellent esterase activity. Finally, two LAB (Lbr9 and Lbr17) and two yeasts (Sf6 and Sc12) were selected for the fermentation test, and the volatile compounds in the final fermented grains were analyzed using HS-SPME-GC-MS.

PCA was conducted on fourteen main flavor compounds in *Baijiu* (Figure 5). By combining the score plot (Figure 3A) and the loading plot (Figure 3B), it can be observed that the fortified fermentation group with Lbr17 exhibited the greatest distance from the origin in the direction of ethyl acetate contribution. Compared to other fortified fermentation groups, the fortified fermentation with Lbr17 displayed the closest distance with CK, indicating that the flavor substances of the two groups were the most similar. Therefore, Lbr17 was finally selected to carry out the fortified inoculation fermentation to study the effect of fortified inoculation with Lbr17 on the fermentation process. *L. brevis* is probably the most important LAB species during fermentation, as it has the highest esterase activity (Figure 1). The esterification of ethanol using certain fungi and the heterolactic fermentation of LAB, ester formation of ethyl acetate, and ethyl lactate were carried out during the fermentation [33]. This was confirmed by our results showing that the concentration of ethyl acetate was higher during fermentation (with Lbr17). Ethyl acetate contributes a fruity (pineapple) and sweet aroma, enhancing the olfactory complexity and bouquet of *Baijiu* [34].

The technique of qPCR can accurately measure the quantity of microorganisms, which has been utilized to study the physiological metabolism and microbial community distribution and applied to the quantitative study of functional bacteria in *Baijiu* fermentation. Li et al. used qPCR to quantify the total bacteria and fungi in the fermentation process of *Fenjiu* [35]. It was found that the content of bacteria was increased while that of fungi was more stable in the fermentation process. Simultaneous investigation of bacterial and fungal variations is important in understanding food fermentation processes.

As shown in Figure 4E, the fortified fermentation group displayed significantly higher levels of *L. brevis* compared to the CK in the initial 15 days of fermentation, indicating that fortified inoculation of *L. brevis* played a significant role during the fermentation process. Figure 4E, along with Figure 4A,C, demonstrated that the increase in *L. brevis* in the fortified fermentation group was significantly higher than that of *Lactobacillus*. Figure 4A revealed that the total number of bacteria was lower than that in the control group after fortified inoculation of Lbr17. These findings suggest that the growth of some *Lactobacillus* and bacteria was inhibited by mutual exclusion while the number of *L. brevis* increased. Figure 4E combined with Figure 4B,D show that the total fungi and yeasts in the fortified fermentation group were lower than those in the control group, indicating that fortified inoculation of *L. brevis* could also inhibit the growth of yeasts and total fungi during fermentation. Therefore, it can be concluded that fortified inoculation of *L. brevis* can selectively modify the microbial communities in the fermentation process via mutual exclusion, thereby improving the control over the fermentation process.

The combined analysis in Figure 5 indicates that fortified fermentation did not alter the overall trend of the fermentation process but instead made fine adjustments. After inoculation on the 15th day, parameters such as pH, titratable acid, moisture content, and reducing sugar tended to stabilize, with the fortified fermentation group displaying higher acid and moisture content compared to the control group. The high acidity of the fortified fermentation group may be due to the acid production of the inoculated *L. brevis* [36]. The higher moisture content in fortified fermentation may be due to more esterification reactions in the fortified fermentation group. Comparing the changes in reducing sugar content in the fortified fermentation group and control group from 1 to 3 days, it was found that fortified inoculation had no significant effect on the formation of reducing sugar.

The higher reducing sugar content in the fortified fermentation group is likely due to decreased enzyme activity, which hampers the process of reducing sugar conversion in ethanol by yeast under low pH conditions (pH stabilizes at about 3.5 after 7 days of fermentation). This reduction in efficiency affects the reducing sugar utilization by certain yeasts. Therefore, the amount of LAB inoculated or mixed with acid-resistant yeast should be properly controlled during inoculation to weaken the influence of low pH on the growth and function of yeast. This ensures that the process of ethanol production by yeast, reducing sugar and acid production by LAB, and reducing sugar and ester production by esterification remains in good dynamic balance. The decrease in ethanol production by yeast a d reducing sugar will result in a decrease in ethanol content or esters in the end product of fermented grains. This study observed a decrease in ethanol content and an increase in esters content, which explains why the alcohol content in fortified fermentation was slightly lower than that in the control group.

The results of Figure 6 indicated that fortified inoculation had the potential to shorten the fermentation time from 28 days to 21 days. According to the physicochemical parameters of the fermentation process, such as moisture content, reducing sugar content, pH, and titratable acid, they tended to remain relatively stable after 15 days. While numerous studies have predominantly concentrated on the exploration of raw materials and functional microorganisms that affect the quality of light-flavor *Baijiu* [3,37], our research offers a prospective and efficacious approach to expedite the fermentation process by studying the changes in the fermentation process of light-flavor *Baijiu* after fortified inoculation of functional microorganisms.

## 5. Conclusions

The physicochemical parameters such as pH, titration acid, moisture content, and reducing sugar content in the fermentation process showed that after inoculation on the first day, the acid content in the fortified fermentation group was significantly higher than that in the control group. Additionally, the moisture content and reducing sugar content in the fortified fermentation group were significantly higher than those in the control group after inoculation on the seventh day. Moreover, the number of *L. brevis* during the fortified inoculation fermentation process was significantly higher than that of the control group 15 days before fortified inoculation. It can be concluded that the metabolism of inoculated *L. brevis* provided more compounds for the formation of *Baijiu* flavor profiles during the fermentation process. Adequate substrate accelerated the microbial kinetics of the fermentation process, improved the efficiency of the fermentation process, and thus shortened the fermentation time. In conclusion, the indigenous *Lactobacillus brevis* isolated from the fermented grain can accelerate the *Baijiu* fermentation process, which provides a theoretical basis for shortening the production cycle of light-flavor *Baijiu* and improving the stability of light-flavor *Baijiu*. In the future, metagenomic and metatranscriptomic techniques can be used to further explore the mechanism of inoculated strains accelerating *Baijiu* fermentation.

## Figures and Tables

**Figure 1 foods-12-04198-f001:**
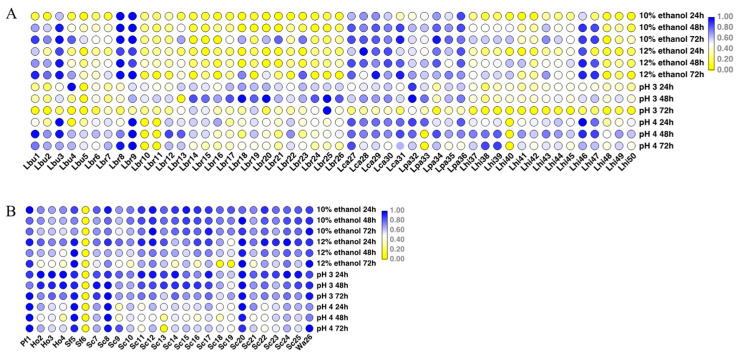
Heatmap of environmental tolerance of bacterial and yeast strains. All data were normalized by row. (**A**) Environmental tolerance of bacteria strains; (**B**) environmental tolerance of yeast strains. *Lactobacillus buchneri* (Lbu), *Lactobacillus brevis* (Lbr), *Lactobacillus casei* (Lca), *Lactobacillus paracasei* (Lpa), *Lactobacillus hilgardii* (Lhi), *Pichia fermentans* (Pf), *Hanseniaspora osmophila* (Ho), *Saccharomycopsis fibuligera* (Sf), *Saccharomyces cerevisiae* (Sc), and *Wickerhamomyces anomalus* (Wa). The label on the lower side represents the strain name, and the label on the right side represents the strain’s culture conditions. The heatmap was drawn after the normalization of OD_600_ values of the strains cultured under different conditions. Yellow indicates that the strain has a small OD_600_ value under this culture condition, and blue indicates that the strain has a large OD_600_ value under this culture condition.

**Figure 2 foods-12-04198-f002:**
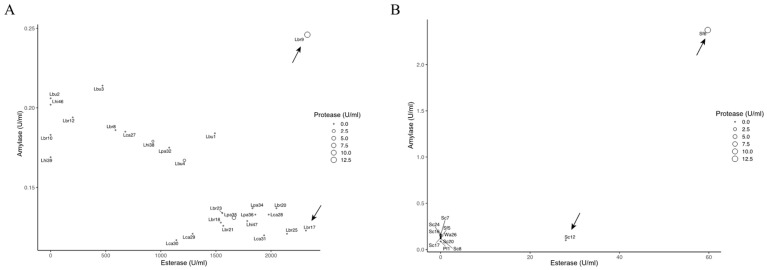
Bubble plot diagram of esterase activity (X-axis), amylase (Y-axis), and protease activity (the bubble size represents protease activity) of selected bacterial strains (**A**); yeast strains (**B**). Arrows indicate the selected strains. Letters indicate the strain name.

**Figure 3 foods-12-04198-f003:**
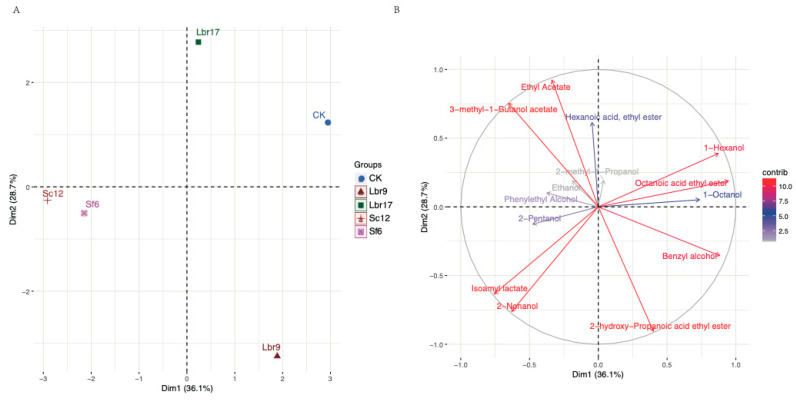
Principal component analysis (PCA) of main volatile compounds of the fortified fermentation of *Baijiu*. (**A**) The PCA score plot for different trials. Dim1 and Dim2 refer to principal component 1 and principal component 2, respectively. Dim1 represents the principal component with the most explanatory data change trends, whereas Dim2 is the principal component with the second most explanatory data change trends. Dots represent individuals, and different colors represent different groups. (**B**) The PCA loading plot shows how strongly each volatile influences the final fortified fermentation. CK: control group fermented with *Daqu* as a starter; Lbr9/Lbr17/Sc12/Sf6: inoculated group with *Daqu* and strain Lbr9/Lbr17/Sc12/Sf6 for fortified fermentation. Different colors indicate the contribution of indicators.

**Figure 4 foods-12-04198-f004:**
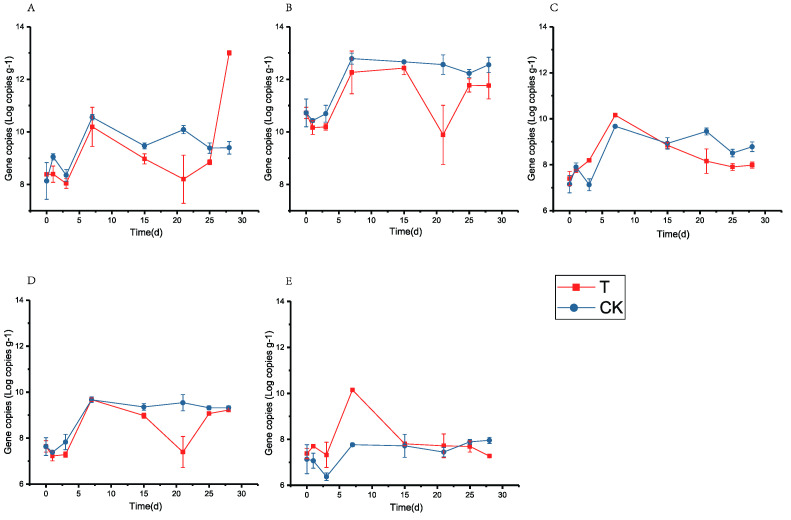
Microbial dynamics in fortified fermentation during fermentation. (**A**) Total bacteria; (**B**) total fungi; (**C**) *Lactobacillus*; (**D**) *Saccharomyces*; (**E**) *L. brevis*; CK: control group fermented with *Daqu* as a starter; T inoculated group with *Daqu* and strain Lbr17 for fortified fermentation. Values represent the means of three replicates with their standard error bars.

**Figure 5 foods-12-04198-f005:**
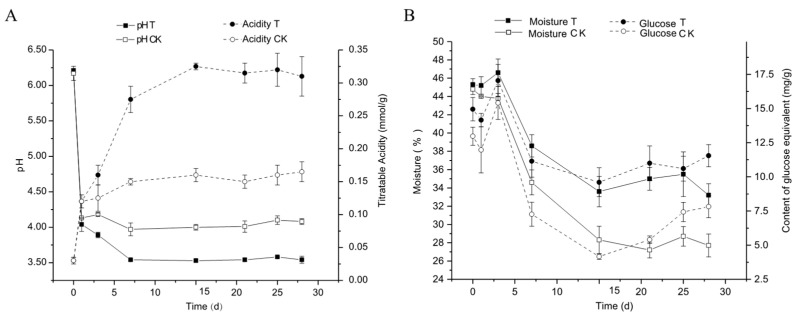
Dynamics of physiochemical properties during the fermentation process. (**A**) The X-axis represents fermentation time, the left Y-axis represents pH value, the right Y-axis represents titrable acidity; (**B**) The X-axis represents fermentation time, the left Y-axis represents moisture the right Y-axis represents reducing sugar. Values represent the means of three replicates with their standard error bars.

**Figure 6 foods-12-04198-f006:**
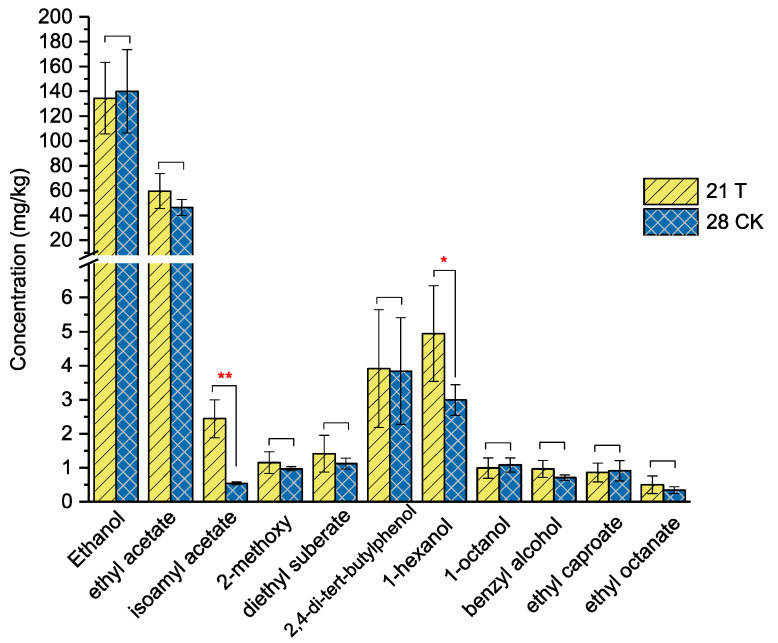
Main volatile compounds concentration after fermentation of 21 d and 28 d. 21T refers to the fermented grains collected after 21 days with *Daqu* and inoculated strain Lbr17; 28CK: control fermented after 28 days only with *Daqu* as a starter. Values represent the means of three replicates with their standard error bars. Values with an asterisk (*) indicate statistical differences between the fortified fermentation group and the control group (*, *p* < 0.05; **, *p* < 0.01).

## Data Availability

The data used to support the findings of this study can be made available by the corresponding author upon request.

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
