# Peer review of "Effect of Fortified Inoculation with Indigenous Lactobacillus brevis on Solid-State Fermentation of Light-Flavor Baijiu"

_foods, 2023, doi:10.3390/foods12234198_

Round 1
Reviewer 1 Report
Comments and Suggestions for Authors
Recommendation: Major
The manuscript Effect of Fortified Inoculation with Indigenous Lactobacillus brevis on Solid-state Fermentation of Light-flavor Baijiu, the methodology was reasonable and technically sound.
Comments to the Author:
The main procedure and findings of the study are well expressed. Introduction: A brief survey of existing literature, the purpose, importance, and innovation of the research is well mentioned. There are major recommendations below
Point 1. Share the important numerical results of the effects of Lactobacillus brevis used in Baijiu beverage in the abtract section.
Point 2. Revise the bibliography instructions for foods journal.
Point 3. Use references for the section below.
Baijiu's flavor profile allows for its classification into three main categories: Sauce-flavor Baijiu, Strong-flavor Baijiu, and Light-flavor Baijiu. Additionally, it can be further classified into nine subgroups: Miscellaneous-flavor Baijiu, Feng-flavor Baijiu, Rice-flavor Baijiu, Medicine-flavor Baijiu, Sesame-flavor Baijiu, Te-flavor Baijiu, Chi-flavor Baijiu, Laobaigan-flavor Baijiu, and Fuyu-flavor Baijiu. Among these categories, the three main categories hold a dominant position in the Baijiu market, collectively accounting for approximately 60% to 70% of the Baijiu consumption in China.
Point 4. You explained your reason for choosing L. brevis from the last sentence of the introduction. However, please add to the last sentence what effects were examined for Baijiu after this election, and what analyzes were made.
Point 5. Write the clear state of the media used in the first use. For example, MRS agar
Point 6. Give your device information used in volatile compound analysis. Explain the method more transitionally.
Point 7. Include group abbreviations for graphics clearly in the descriptions.
Point 8. Explain the p significance levels of * for Figure 6.
Point 9. It is not appropriate to give table figures under a separate heading. Please read the sample trowels and magazine instructions in foods magazine in detail and edit the manuscript again.
Point 10. Interpretation of PCA analysis results is insufficient. Explain further the links of volatile compounds between groups.
Point 11. The details on primer sequences and thermal cycles were given in Table S2. ????
I did not see any tables in the article. Review the tables again.
Point 12. There are quality studies on Baijiu made in 2023. The authors agreed that they did not benefit from this study. I suggest that they add current studies done in 2023 to the article.
Point 13. Make suggestions for future studies in the conclusion part.
Comments on the Quality of English LanguageThere are some minor grammatical errors.
Author Response
Thank you for your professional comments. All the revisions in the revised manuscript are shown using red highlights. The point-to-point answers and explanations for all revisions were listed in a separate file following this letter.

Reviewer 2 Report
Comments and Suggestions for Authors
The manuscript entitled Effect of Fortified Inoculation with Indigenous Lactobacillus brevis on Solid-state Fermentation of Light-flavor Baijiu present s information related to use of L. brevis to improve different characteristics of Baijiu. The manuscript presents critical issues that authors must attend to. Below are the comments.
-Line 31. Te-flavor? Did you mean tea-flavor?
-What was the objective of the work? It is not clear in the manuscript
-Provide the name and geographical coordinates for the place where the grain samples were obtained.
-Line 88. What was the solid-liquid ratio of diluted samples?
- Line 98. Deposit the DNA sequences of the microorganisms in a data bank such as GenBank. Provide the accession numbers.
- Provide the details for the enzyme quantification methods. How the enzymatic units were defined?
-Section 2.4. What was the rationale for presenting this section? It is not clear what was evaluated. Instead of this section, the authors must depict the details for the solid-state fermentation.
- Section 2.5. Provide the details for the GC-MS methods.
-Materials and methods section. How was carried out the solid-state fermentation? In the title and the objectives, it is mentioned the authors evaluated a SSF.
- Section 3.5. What was the rationale for this section? The figures must be presented right after the first time they are mentioned in the manuscript.
- The presentation of the results must be improved. The expressed ideas are not clear.
Comments on the Quality of English LanguageEnglish language is ok.
Author Response
-Line 31. Te-flavor? Did you mean tea-flavor?
Thank you for your professional comments. However, this is indeed Te-flavor. “Te” is Chinese pinyin, representing "Si Te ", originated in ancient China's "Si Te " baijiu, so it is named Te-flavor baijiu.
-What was the objective of the work? It is not clear in the manuscript
Thank you for your professional comments. In order to improve the stability of Baijiu quality, it is an effective method to add specific starter culture, and the indigenous strains isolated from the original environment have high adaptability and particularity. Therefore, in this study, excellent strains were first isolated from Fermented grain samples, and finally, their potential to promote Baijiu solid fermentation was analyzed on a laboratory scale. To make it clear, we have added the objective of our work in introduction section, please see line 90-98.
-Provide the name and geographical coordinates for the place where the grain samples were obtained.
Thank you for your professional comments. We have added the place where the grain samples were obtained in section 2.1 ( Line 102).
-Line 88. What was the solid-liquid ratio of diluted samples?
Ten grams of fermented grains were suspended in 90 mL of phosphate-buffered saline (PBS) solution, followed by 10-fold serial dilutions. Subsequently, four specific dilutions, namely -5, -6, -7, and -8, were selected for spreading onto agar plates. Further elucidation has been included in line 103-106 for enhanced clarity.
- Line 98. Deposit the DNA sequences of the microorganisms in a data bank such as GenBank. Provide the accession numbers.
We have already deposit the DNA sequences in GenBank(OK35462-OK354403).
- Provide the details for the enzyme quantification methods. How the enzymatic units were defined?
Thank you for your professional comments. We have added the enzyme quantification methods. Please see line 131-137
-Section 2.4. What was the rationale for presenting this section? It is not clear what was evaluated. Instead of this section, the authors must depict the details for the solid-state fermentation.
Thank you for your suggestion. Here, in the section 2.4, we would like to firstly show the candidate strains for fortified fermentation and their growth condition. After that, we would like to introduce the method of fortified fermentation, according to the production process for light-flavor Baijiu. We have added the specific steps of solid-state fermentation process. Please see line 145-153.
- Section 2.5. Provide the details for the GC-MS methods.
Thank you for your professional comments. We have supplemented the device information used in volatile compound analysis. Please see line 160-162.
- Materials and methods section. How was carried out the solid-state fermentation? In the title and the objectives, it is mentioned the authors evaluated a SSF.
We have already added the description and reference for solid-state fermentation procedures. Please see line 145-152.
- Section 3.5. What was the rationale for this section? The figures must be presented right after the first time they are mentioned in the manuscript.
Thank you for your professional comments. We have rearranged the position of the figures.
- The presentation of the results must be improved. The expressed ideas are not clear.
We have already carefully modified the presentation in the results section.

Reviewer 3 Report
Comments and Suggestions for Authors
I revised the manuscript. I suggested some corrections and I aded citation for methods on the manuscript.
Manuscript mus be revised for English grammer by authors.

Comments on the Quality of English LanguageI revised the manuscript. I suggested some corrections and I aded citation for methods on the manuscript.
Manuscript mus be revised for English grammer by authors.
Author Response
Thank you for providing valuable feedback on the manuscript. We have made revisions to address the comments in the revised version of the manuscript. Furthermore, we have thoroughly reviewed your work and found it highly beneficial, incorporating relevant citations from it into the text. We sincerely appreciate your efforts and suggestions in improving this article.

Reviewer 4 Report
Comments and Suggestions for Authors
The manuscript "Effect of Fortified Inoculation with Indigenous Lactobacillus brevis on Solid-state Fermentation of Light-flavor Baijiu" describes an iteresting study about a specific solid state fermented product and its fermentation dinamic in term of microbial population. Selected culture have also been tested and microbila dinamic and volatile components profile have reported to describe specific features of this fermented product. The study is intersting and Introduction, Methods, and Results and well described. In the Results a more clear description about the significance of differences reported for the group "Total bacteria" "Lactobacillus" and L.brevis" should be added. The Discussion should be improved reporting similarity and differences from other studies (and related references).
Few other detail can be improved as follow:
Line 19: please change "the inoculated of L. brevis" with "the inoculated L. brevis"
Line 31: change M with m
Line 112: please clarify the menaing of PNP
Line 140: please add "spp" after Lactobacillus and Saccharomyces to identify the genus specificity and change accordingly along the whole manuscript
Line 141: please add L. brevis before the strain code
Lines 161-163. use abbreviation consistently (i.e L. instead of Lactobacillus) and check for similar issue along the whole manuscript
Line 172-173: use abbreviation consistently (i.e Sa. instead of Saccharomycopsis) and check for similar issue along the whole manuscript
Line 205-206: Please clarify this statement
Line 247 L. brevis must be in Italic
In 3.5. Figures, Tables and Schemes, since all figures, tables and schemes must be self explanatory add a complete description of abbreviations and labes in the respective caption, number of replicates, meaning of the error bars etc etc.
Comments on the Quality of English LanguageA mother tongue English speaker may help for minor changes
Author Response
Line 19: please change "the inoculated of L. brevis" with "the inoculated L. brevis"
Thank you for your professional comments. We have changed the word.
Line 31: change M with m
Thank you for your professional comments. We have changed the letter.
Line 112: please clarify the menaing of PNP
Thank you for your professional comments. We have added the full name of the word.
Line 140: please add "spp" after Lactobacillus and Saccharomyces to identify the genus specificity and change accordingly along the whole manuscript
Thank you for your professional comments. We have added the word.
Line 141: please add L. brevis before the strain code
Thank you for your professional comments. We have added the word.
Lines 161-163. use abbreviation consistently (i.e L. instead of Lactobacillus) and check for similar issue along the whole manuscript
Thank you for your professional comments. We have changed the abbreviation and checked the whole manuscript.
Line 172-173: use abbreviation consistently (i.e Sa. instead of Saccharomycopsis) and check for similar issue along the whole manuscript
Thank you for your professional comments. We have changed the abbreviation and checked the whole manuscript.
Line 205-206: Please clarify this statement
Thank you for your professional comments. We have clarified this statement.
Line 247 L. brevis must be in Italic
Thank you for your professional comments. We have italicize the word.
In 3.5. Figures, Tables and Schemes, since all figures, tables and schemes must be self explanatory add a complete description of abbreviations and labes in the respective caption, number of replicates, meaning of the error bars etc etc.
Thank you for your professional comments. We have added the information of the tables and figures.
Round 2
Reviewer 1 Report
Comments and Suggestions for Authors
The authors made the necessary corrections.
Author Response
Thank you for your reply.
Reviewer 2 Report
Comments and Suggestions for Authors
The authors have addressed almost all the comments. I just have a minor comment.
-Provide the details for the elution method used in the GC-MS separation.
Comments on the Quality of English LanguageI have no comments.
Author Response
Thank you for your professional comments. We added some details about the GC-MS method. Please see line 159-164.

Reviewer 4 Report
Comments and Suggestions for Authors
In the revised version of the manuscription "Effect of Fortified Inoculation with Indigenous Lactobacillus brevis on Solid-state Fermentation of Light-flavor Baijiu" there is still a little need for improvement in relation to the figure description, some part of the discussion and conclusion. The specific details are described in the comments reported in the attached file.

Comments on the Quality of English Languageas above
Author Response
感谢您的专业评论。修订稿中的所有修订均使用红色突出显示。
